# Bullying at School, Cyberbullying, and Loneliness: National Representative Study of Adolescents in Denmark

**DOI:** 10.3390/ijerph21040414

**Published:** 2024-03-28

**Authors:** Katrine Rich Madsen, Mogens Trab Damsgaard, Kimberly Petersen, Pamela Qualter, Bjørn E. Holstein

**Affiliations:** 1National Institute of Public Health, University of Southern Denmark, 1455 Copenhagen, Denmark; krma@sdu.dk (K.R.M.); trab@sdu.dk (M.T.D.); 2School of Education, University of Leeds, Leeds LS2 3AR, UK; k.petersen@leeds.ac.uk; 3Manchester Institute of Education, University of Manchester, Manchester M1 5AN, UK; pamela.qualter@manchester.ac.uk

**Keywords:** adolescents, bullying, victimization, cyberbullying, HBSC, loneliness

## Abstract

Aims: The aim was to examine how loneliness was associated with bullying victimization at school and online. Methods: We used data from the Danish arm of the international Health Behavior in School-aged Children (HBSC) study from 2022. The study population was a nationally representative sample of 11–15-year-olds who completed the internationally standardized HBSC questionnaire at school, *n* = 5382. Multilevel logistic regression was applied to study the associations between bullying victimization and loneliness. Results: The prevalence of reporting loneliness often or very often was 9.0%; 6.3% of the sample experienced habitual bullying victimization at school, and 4.8% incurred cyberbullying. There was a strong and graded association between loneliness and bullying victimization at school and cyberbullying. The associations were significant for boys and girls, and the association between exposure to bullying at school and loneliness was steeper for boys than girls. The gradients were steeper for physical bullying than for cyberbullying. Students exposed to habitual bullying in both contexts had an adjusted OR (95% CI) of 11.21 (6.99–17.98) for loneliness. Conclusion: Exposure to bullying at school and cyberbullying are strongly associated with loneliness. It is important to reduce bullying at school and on the internet and to promote effective interventions to reduce continuing loneliness.

## 1. Introduction

Loneliness is a subjective feeling of isolation. It is often defined as a cognitive discrepancy between the social relations an individual wishes to have and those that one perceives to have, and the affective reactions of sadness and emptiness that follow [1]. The feeling of loneliness is common in adolescence [2,3,4,5], and many adolescents will experience loneliness for short periods. The reasons may be feeling left out among peers, a change of school, parental divorce, or other adverse life events [6,7]. However, some adolescents experience prolonged feelings of loneliness that result from repeated failure to reconnect with others, which is a serious threat to their quality of life [6,7,8] and academic performance [9]. A recent meta-analysis of longitudinal studies suggested that loneliness tended to remain stable from adolescence to adulthood [10]. Loneliness is also an important public health problem because it is associated with a range of health problems [1,6,11,12,13,14] and risk behaviors [15,16,17]. It is important to understand the precursors of loneliness to strengthen preventive efforts. The current study focuses on two potential precursors: bullying victimization at school and exposure to cyberbullying.

Bullying victimization at school is common among adolescents [2,18,19], although the prevalence has been diminishing over the past decades in Europe and North America [20,21]. There is abundant documentation for an association between exposure to bullying and adverse psychological consequences such as poor life satisfaction [18], mental health problems, and suicidal behavior [5,22,23,24,25,26,27,28,29]. A small number of cross-sectional [30,31,32,33,34,35] and prospective [35,36,37] studies confirm that there is an association between loneliness and bullying victimization at school. For instance, Due et al. [30] found a strong and graded association between loneliness and exposure to bullying at school, a finding which was consistent across twenty-eight countries. The effect sizes vary across studies, from small to large. The variation in effect sizes suggests a need for further studies.

Exposure to cyberbullying, sometimes labeled internet bullying, online victimization, or internet harassment, is the use of digital technologies to harass, threaten, embarrass, or target another person. This phenomenon is also common among adolescents [2,18,21,24,38,39], although the prevalence of exposure to cyberbullying is lower than exposure to bullying at school [21]. The studies that find an association between exposure to cyberbullying and loneliness [40,41,42] show considerable variations in effect sizes, from weak to strong. There is some doubt about the causal pathway since a few prospective studies show that loneliness is a precursor of cyberbullying rather than the reverse [43]. As with face-to-face, in-person bullying victimization, the variation in effect sizes across studies for the association between loneliness and cyberbullying highlights the need for further studies.

There are reasons why there might be differences in the associations between loneliness and face-to-face, in-person bullying and loneliness and cyberbullying. For example, Van den Eijnden et al., 2014 [44], emphasize that findings from research on bullying at school cannot automatically be transferred to cyberbullying because these two phenomena differ in important ways: Cyberbullying via the internet has much higher accessibility to the target than bullying during school hours. Cyberbullying can reach a much larger audience than bullying at school and may remain visible for a long time to the victim and the audience, potentially resulting in longer-lasting negative effects, e.g., loneliness. A study covering six North European countries showed little overlap between bullying at school and cyberbullying, suggesting that the two may be different phenomena [18]. It is, therefore, important to analyze which kind of exposure is more closely associated with loneliness and to analyze the association between loneliness and double exposure (to bullying at school and cyberbullying). Only a few studies focus on such combined effects, and they found higher rates of loneliness among adolescents who were exposed to bullying in both contexts [31]. Van den Eijnden et al. [44] suggest that the two kinds of bullying are mutually reinforcing, i.e., exposure to bullying in one context increases the risk of exposure in the other. Studies about the association between exposure to bullying and loneliness use different reference periods, which makes comparisons difficult [40]. A few studies suggest that the association between loneliness and exposure to bullying varies by sex and age group [33,43,45]. According to Cava et al. [45], older teenage girls might be more vulnerable to cyberbullying than boys, for instance when exposed to cyber-control from a romantic relationship. These girls reported more feelings of loneliness and assessed their social network as worse than those never victimized. Landstedt and Persson [24] suggest more focus on the gender issue.

There is a need for further exploration of the association between loneliness and exposure to bullying, which includes both kinds of bullying and uses identical reference periods for the measurement of exposure. The aim of this study was to examine how loneliness was associated with exposure to bullying at school, to cyberbullying, and combinations of bullying at school and cyberbullying.

## 2. Methods

### 2.1. The Study

We collected data for the Danish arm [46] of the international Health Behavior in School-aged Children (HBSC) study [2,47]. This cross-sectional school survey conducted in 2022 included a nationally representative sample of three age groups: 11-, 13-, and 15-year-olds. Inclusion process: We selected at random a sample of schools from a complete list of public and private schools in Denmark but excluding schools for children with learning disorders. In each participating school, we invited all fifth-, seventh-, and ninth-grade students (corresponding to the three age groups) to participate and complete the internationally standardized HBSC questionnaire in the classroom [48]. Inclusion criteria: All students in the relevant classes in the participating schools. The participation rate among students, calculated as a percentage of students enrolled in the participating classes who completed the questionnaire, was 69.0%, *n* = 5823. Exclusion criteria: students absent from school at the day of data collection, students who didn’t want to participate, students whose parents rejected their children’s participation, students with missing data about loneliness, exposure to bullying, and applied control variables. The final study population was 5382 (92.4% of eligible students).

### 2.2. Measurements

The study used one item for the measurement of loneliness: “Do you feel lonely?” (never, sometimes, often, or very often). We dichotomized the responses into no (never or sometimes) and yes (often or very often) to separate students with prolonged feelings of loneliness from students with less severe and more transient feelings of loneliness or a complete absence. This single-item measure and the more elaborated University of California Los Angeles (UCLA) Loneliness Scale showed remarkably similar patterns of association with health, sleep, and scholastic self-beliefs [6,9]. Mund et al. [49] showed that such a direct single-item measure correlated highly with other measures of loneliness. Furthermore, interviews with adolescents about their understanding and perceptions of loneliness indicate that this question has good face validity [50]. These findings suggest that the measure is valid for the purpose of our study.

We measured exposure to bullying at school using the item “Bullying is repeated episodes of being kept outside, teased, beaten, or bothered in ugly ways. How often have you been bullied at school in the past couple of months?” with the response options (1) “I have not been bullied at school in the past couple of months”, (2) “It has only happened once or twice”, (3) “Two to three times a month”, (4) “About once a week”, and (5) “Several times a week”. Students who did not answer the question (*n* = 130) were included in the non-exposed category. Kyriakides et al. [51] showed that students’ reports about bullying victimization to bullying at school were trustworthy. The measurement of cyberbullying used the item “In the past couple of months how often have you been cyberbullied (e.g., someone sent mean instant messages, email, or text messages about you; wall postings; created a website making fun of you; posted unflattering or inappropriate pictures of you online without permission or shared them with others)?” with the same response options and with 138 students who did not answer the question included in the non-exposed category. The descriptive analyses separated habitual exposure (response options 3–5, at least two to three times a month) from less exposure (response options 1–2) because it is habitual bullying that has severe consequences for future mental health [26]. Finally, we constructed a combined measure of exposure to habitual bullying at school and cyberbullying with four categories: (1) not exposed to any bullying, (2) exposed to cyberbullying but not bullying at school, (3) exposed to bullying at school but not cyberbullying, and (4) exposed to both kinds of bullying.

The analyses included four socio-demographic control variables: sex; age group (11-, 13-, and 15-year-olds); origin (native Danish, descendants of immigrants, immigrants, 24 unclassifiable students excluded), based on items about the student’s, the father’s, and mother’s country of birth; and socioeconomic status. Socioeconomic status was measured via eight items: “Does your father/mother have a job?”, “If no, why does he/she not have a job?”, “If yes, please say in what place he/she works (for example, hospital, bank, or restaurant)”, and “Please write down exactly what job he/she does there (for example, teacher or bus driver)”. The research group coded the answers in accordance with the Danish Occupational Social Class (OSC) measure from I (high) to V (low) [52]. We added OSC VI for economically inactive parents who received unemployment benefits, disability pensions, or other transfer income, similarly based on students’ responses. Most students (81.6%) provided sufficient information for the coding of OSC, and students with insufficient information were labeled “unclassifiable.” Schoolchildren in these age categories can report their parents’ occupation with a high agreement with parents’ own information [53,54,55,56]. Pförtner et al. [57] showed that OSC is an appropriate variable for studies of social inequality in adolescents’ health. Each participant was categorized using the highest-ranking parent into four levels of family OSC: high (I-II, e.g., professionals and managerial positions), middle (III-IV, e.g., technical and administrative staff and skilled workers), unclassifiable, and low (V, unskilled workers, and VI, economically inactive).

### 2.3. Ethics Approval and Informed Consent

The study complied with national legislation about ethical approval, consent, and data protection. The study was approved by the Research Ethics Committee at the University of Southern Denmark. We asked the school board, which is the parents’ representative, the principal, and the students’ council in each of the participating schools, to approve the study. Prior to the study, the parents received an electronic link to a short video with information about the study and assurance that participation was voluntary and confidential. The parents also received an electronic link by which they could reject their child’s participation in the study. Prior to the data collection, the students viewed a short video providing information about the study, emphasizing that participation was voluntary and confidential. The applied data file was pseudonymized, i.e., did not include information about the participating schools or participants such as name, address, or data of birth.

### 2.4. Statistical Procedures

We used SAS version 9.4 for the analyses (SAS Institute Inc., Cary, NC, USA). The first step was crosstabulations and the use of the chi^2^-test for homogeneity. The second step was multilevel (PROC GLIMMIX) logistic regression analyses to examine the associations between loneliness and the exposure variables, adjusted for sex, age group, origin, and OSC. We also conducted stratified logistic regression analyses to examine whether the pattern of associations was similar for boys and girls and in the three age groups.

## 3. Results

The overall prevalence of loneliness was 9.0%. Table 1 shows that loneliness was significantly more prevalent among girls vs. boys, 15-year-olds vs. 11-year-olds, immigrants vs. Danish origin, and students from lower vs. higher OSC. The proportion exposed to bullying at school at least a couple of times per month was 6.3%, significantly more prevalent among girls vs. boys, 11-year-olds vs. 15-year-olds, immigrants vs. Danish origin, and students from lower vs. higher OSC. The proportion exposed to cyberbullying at least a couple of times per month was 4.8% and not significantly related to any of the socio-demographic variables. The variable that combined exposure to habitual bullying at school and cyberbullying classified the students into four categories: 4867 (90.4%) were not exposed to any bullying, 175 (3.3%) were exposed to cyberbullying but not bullying at school, 259 (4.8%) were exposed to bullying at school but not cyberbullying, and 81 (1.5%) were exposed to both kinds of bullying. These figures suggest that there is little overlap between exposure to the two types of bullying since most of the students who were exposed to bullying in one context were unexposed in the other context.

Table 2 shows a strong and graded association between loneliness and exposure to bullying. Even among students with low exposure to bullying (once or twice in the past couple of months), the odds ratio (OR) for loneliness was significantly elevated compared to non-exposed students. The OR (95% CI) for loneliness was 11.58 (7.21–18.61) among students exposed to bullying at school several times a week. The corresponding figure for exposure to cyberbullying was 5.79 (3.37–9.86). Finally, the OR for loneliness among the few students exposed to both kinds of bullying was 10.80 (6.87–16.97). Table 2 also shows that the OR estimates changed little when adjusted for socio-demographic control variables. Separate analyses for boys and girls and the three age groups showed a significant association between loneliness and the three measures of bullying in every sub-group (not shown in the table). The association between exposure to bullying at school and loneliness was steeper for boys than girls, manifested via a statistically significant interaction term between sex and exposure to bullying at school, *p* = 0.0165.

## 4. Discussion

### 4.1. Key Findings

The prevalences of loneliness and exposure to bullying resemble findings from other recent studies of adolescents from Northwestern Europe [2,3,6,19,20,58]. The key finding was the strong and graded association between loneliness and exposure to bullying in both contexts, at school and online, even after adjustment for sex, age group, origin, and socioeconomic status. Students exposed to bullying in both contexts had around ten times higher OR for loneliness. Even a low exposure to bullying more than doubled the likelihood of loneliness. These observations correspond with other studies [30,31,32,33,35,36,37], but the association between bullying and loneliness was extraordinarily strong in our study. The finding is also consistent with the many studies that document a strong association between exposure to bullying and a broad range of mental health problems [5,22,23,25,26,27,28,29,30,59].

The significant and graded association between loneliness and exposure to bullying appeared among boys and girls and in all three age groups. The association between bullying at school and loneliness was significantly steeper for boys than girls. We have not been able to identify studies which confirm this difference between boys and girls. We were surprised to see the modest overlap between exposure to bullying at school and cyberbullying. The two kinds of bullying showed different patterns of association with socio-demographic variables. These findings suggest that the two kinds of bullying are qualitatively different phenomena.

While it seems reasonable that exposure to bullying might increase the feeling of loneliness, it remains to be determined whether bullying is solely a direct precursor for loneliness or whether bullying victimization and loneliness are both determined by some other contextual factors, e.g., the school environment. Bullying should be understood as a behavior deeply rooted in the school’s socio-environmental context. For instance, negative school perceptions among students [60] and staff [61] and a school environment where teachers ignore or dismiss bullying [62] are strongly associated with bullying. Further, a negative school environment, where students perceive their peers and teachers as unsupportive, is strongly associated with loneliness [34]. Therefore, it is likely that some of the association between bullying and loneliness is due to a common causal factor: a negative school environment. Other contextual factors such as a gendered social life [24] and socioeconomic background [4,58] may also play a role as contextual factors shaping the room for bullying victimization and loneliness.

### 4.2. Methodological Issues

The study’s strength is the large and nationally representative study population and the robustness of the applied measurements. Furthermore, the reference time for measuring exposure to bullying in the two contexts was similar: the last couple of months. There are important limitations as well. One is the cross-sectional design, which limits insight into causality. This limitation is particularly important because there are indications of opposite pathways: some longitudinal studies show that exposure to bullying at school predicts loneliness [36,37], while other studies suggest that loneliness predicts cyberbullying [43,44].

The study may suffer from selection bias. The participation rate among pupils was high (69.0%), but it is likely that students who were frequently bullied and/or felt lonely were more absent from school and, therefore, did not participate in the study. This could result in an underestimation of the prevalence of exposure to bullying and the prevalence of loneliness among adolescents. If there is an underestimation of both exposure and outcome, the analyses could potentially also underestimate the association between exposure to bullying and loneliness.

The applied measurement of loneliness requires the individual to identify and label him- or herself as lonely, which may be perceived as a social stigma [6,50]. This may be why our study identified fewer lonely adolescents than studies using multi-item measures such as the UCLA scale, which does not mention the term lonely. Eccles et al. [6] found a high correlation between the one-item measure and the multi-item UCLA scale. Therefore, we do not expect that our single-item measure of loneliness invalidates the finding of an association between loneliness and exposure to bullying.

The study was conducted in 2022, just after the close-down of the society due to the COVID19 pandemic. It is very likely that the prevalence of loneliness and the prevalence of bullying was higher in this period than the years before [63,64] and that may have influenced the findings.

### 4.3. Implications

From a research point of view, it is important to seek information from prospective studies. It is still an open question whether loneliness predicts cyberbullying or the other way around. The long-term adverse consequences of bullying at school are serious and well-documented [5,22,23,24,25,26,27,28,29], whereas we need more insight into the long-term consequences of cyberbullying. Another priority is to learn more about factors that modify the associations between loneliness and exposure to bullying because modifying factors may guide future intervention programs. A potential modifier is social capital at school [65]. Schnepf et al. 2023 [34] suggest that we need more information about the role of the school environment in generating loneliness. They also suggest we need to know more about what schools can do to protect adolescents from loneliness. Furthermore, social contexts such as gendered social life and socioeconomic background could play a role as drivers of loneliness and drivers of bullying. This aspect needs to be studied carefully.

Although there is some documentation that the measure on bullying is trustworthy [51], we need qualitative studies to dig deeper into what this concept means among adolescents. Moreover, we would like to see studies which expand the perspective from exposure to bullying to other kinds of exposure to violence at school, e. g., being a witness to bullying and being involved in physical fighting. Finally, there is a need for studies which examine if all kinds of exposure to violence at school are drivers of loneliness.

The findings suggest that interventions against loneliness in adolescence are needed. Such interventions are feasible and often effective [66]. Interventions in the school environment are a way to decrease loneliness, particularly interventions that promote a more cooperative climate between students and improve teachers’ support for their students [34]. The findings also underline the importance of implementing bullying prevention interventions at school [67,68,69]. There is less insight into interventions against cyberbullying. According to van den Eijnden et al. 2014 [44], it seems crucially important to teach adolescents and parents about the risks of cyberbullying and to provide them with tools on how to interpret and deal with such experiences. Cosma et al. [21] likewise suggest that we need a more holistic perspective on public health programs and policies that address bullying more broadly rather than focusing on behaviors that happen in a particular context.

## 5. Conclusions

Exposure to bullying at school and cyberbullying are strongly associated with loneliness. It is important to reduce bullying at school and on the internet and to promote effective interventions to reduce continuing loneliness. The main limitations of the study are the cross-sectional design which limits the insight into causality and the potential selection bias that may result in an underestimation of loneliness and bullying and probably also of the association between the two. The study suggests that there is a need for interventions against loneliness and against bullying among adolescents. The school is an ideal setting for interventions as it is a breeding place for both bullying and loneliness. It is possible to target the entire adolescent population in the school setting, and there is solid documentation for the effect of interventions against bullying at school.

## Figures and Tables

**Table 1 ijerph-21-00414-t001:** Loneliness and exposure to bullying at school and cyberbullying due to socio-demographic variables.

	Pct. Lonely Often or Very Often	Pct. Exposed to Bullying at School ^a^	Pct. Exposed to Cyberbullying ^a^
Sex			
Boys (*n* = 2659)	5.5	5.3	3.5
Girls (*n* = 2723)	12.4 *	7.1 *	4.1
Age groups			
11-year-olds (*n* = 1896)	7.0	7.3	4.9
13-year-olds (*n* = 1943)	10.1	6.9	5.3
15-year-olds (*n* = 1543)	10.0 *	4.3 *	4.0
Origin			
Native Danish (*n* = 4857)	8.6	6.1	4.7
Descendants of immigrants (*n* = 325)	10.8	7.7	4.3
Immigrants (*n* = 200)	16.0 *	10.5 *	7.0
Occupational Social Class			
High (*n* = 2369)	7.7	5.4	4.1
Medium (*n* = 1717)	8.8	6.6	5.0
Unclassifiable (*n* = 849)	10.7	6.1	5.3
Low (*n* = 447)	13.4 *	10.5 *	6.7
Total (*n* = 5382)	9.0	6.3	4.8

^a^ Defined as habitual bullying, i.e., at least two to three times per month * *p* < 0.01.

**Table 2 ijerph-21-00414-t002:** OR (95% CI) for loneliness often or very often by exposure to bullying and covariates.

	Pct. Lonely	Crude OR (95% CI)	Adjusted OR (95% CI)
Sex			
Boys (*n* = 2659)	5.5	1 (ref.)	1 (ref.)
Girls (*n* = 2723)	12.4	2.41 (1.97–2.95)	2.47 (2.02–3.03) ^a^
Age groups			
11-year-olds (*n* = 1896)	7.0	1 (ref.)	1 (ref.)
13-year-olds (*n* = 1943)	10.1	1.52 (1.21–1.93)	1.62 (1.28–2.05) ^a^
15-year-olds (*n* = 1543)	10.0	1.50 (1.17–1.92)	1.59 (1.24–2.04) ^a^
Origin			
Native Danish (*n* = 4857)	8.6	1 (ref.)	1 (ref.)
Descendants of immigrants (*n* = 325)	10.8	1.23 (0.84–1.78)	1.11 (0.76–1.64) ^a^
Immigrants (*n* = 200)	16.0	1.98 (1.32–2.91)	1.70 (1.13–2.56) ^a^
Occupational Social Class			
High (*n* = 2369)	7.7	1 (ref.)	1 (ref.)
Medium (*n* = 1717)	8.8	1.16 (0.93–1.46)	1.17 (0.93–1.47) ^a^
Unclassifiable (*n* = 849)	10.7	1.44 (1.10–1.88)	1.56 (1.19–2.06) ^a^
Low (*n* = 447)	13.4	1.81 (1.32–2.49)	1.77 (1.27–2.46) ^a^
Bullied at school:			
Not bullied (*n* = 4373)	6.0	1 (ref.)	1 (ref.)
Once or twice (*n* = 669)	15.8	2.92 (2.29–3.72)	3.11 (2.42–3.99) ^b^
Two to three times a month (*n* = 156)	29.5	6.37 (4.41–9.20)	6.46 (4.41–9.45) ^b^
About once a week (*n* = 109	33.0	7.63 (5.01–11.62)	8.10 (5.24–12.51) ^b^
Several times a week (*n* = 75)	42.7	11.34 (7.04–18.27)	11.37 (6.94–18.64) ^b^
Cyberbullied:			
Not bullied (*n* = 4541)	6.9	1 (ref.)	1 (ref.)
Once or twice (*n* = 585)	17.4	2.84 (2.22–3.62)	2.75 (2.14–3.54) ^b^
Two to three times a month (*n* = 130)	26.2	4.65 (3.08–7.01)	4.79 (3.13–7.32) ^b^
About once a week (*n* = 59)	27.1	5.06 (2.81–9.12)	5.44 (2.97–9.98) ^b^
Several times a week (*n* = 67)	29.9	5.79 (3.38–9.94)	7.65 (4.37–13.39) ^b^
Combined measure:			
Not habitually bullied (*n* = 4867)	6.9	1 (ref.)	1 (ref.)
Only cyberbullied (*n* = 175) ^c^	19.7	3.21 (2.17–4.76)	3.67 (2.45–5.50) ^b^
Only bullied at school (*n* = 259) ^c^	30.1	5.68 (4.26–7.59)	5.66 (4.20–7.64) ^b^
Both kinds of bullying (*n* = 81) ^c^	44.4	10.63 (6.74–16.75)	11.21 (6.99–17.98) ^b^

^a^ Sex, age group, origin, and occupational social class mutually adjusted. ^b^ Adjusted for sex, age group, origin, and occupational social class. ^c^ Defined as habitual bullying, i.e., at least two to three times per month.

## Data Availability

Applications to access the dataset should be sent to the Primary Investigator of the Danish HBSC Study, Dr. Katrine Rich Madsen, krma@sdu.dk.

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
