# Peer review of "Bullying at School, Cyberbullying, and Loneliness: National Representative Study of Adolescents in Denmark"

_ijerph, 2024, doi:10.3390/ijerph21040414_

Round 1

Reviewer 1 Report

Comments and Suggestions for Authors

Thank you for inviting me to review this manuscript which I found very interesting. I just have a few points that could help to strengthen it. I hope these comments are useful.

Typo on the last line of the abstract and the manuscript: “at the internet’ should be ‘on the internet’.

Introduction

Can you say anything about the pandemic? Given that data were collected in 2022 when restrictions were coming to an end is it possible that young people might have felt loneliness related to bullying, particularly online, at this time?

Method

Although the HBSC data was used for this study, did you the authors collect this original data too? (please see end of page 2 under ‘The study’).

Can you reflect on the ethical implications of the questionnaire? How was consent sought? How were data stored etc? What options were provided to students who did not want to participate?

Interesting that cyberbullying was defined/explained for the students. Was a definition of ‘loneliness’ or ‘bullying’ provided?

Discussion

The discussion provides links with your study and the wider literature, but an exploration of the nuanced findings would add depth to the section. For example, exploring the gendered and socio-economic aspects behind loneliness or native versus non-native student’s experiences and the two types of bullying. It is possible that loneliness is related to the school environment, but these factors could also play a part.  It would be beneficial to draw on the unique results from your study rather than focusing on what they support in the literature.

Implications:

The paper begins with a sentence stating that bullying is subjective, have you considered the idea of research with young people to determine their understanding of this term? As you say early in the manuscript, loneliness is subjective. Therefore, does future exploration of this topic lend itself to a qualitative study exploring this issue?

Conclusion

More information is needed in the conclusion which is two sentences. What are the takeaway messages for this research? 

Author Response

Reviewer 1

Thank you for inviting me to review this manuscript which I found very interesting. I just have a few points that could help to strengthen it. I hope these comments are useful.

Response: Thank you for your kind words. Your comments are much appreciated.

Typo on the last line of the abstract and the manuscript: “at the internet’ should be ‘on the internet’.

Response: Now corrected – We used an English editing service to scrutinize the manuscript.

Introduction

Can you say anything about the pandemic? Given that data were collected in 2022 when restrictions were coming to an end is it possible that young people might have felt loneliness related to bullying, particularly online, at this time?

Response: There is not much evidence about how the pandemic influenced bullying and loneliness among adolescents, but we have mentioned the issue and included two new references about this issue in the Discussion section.

Method

Although the HBSC data was used for this study, did you the authors collect this original data too? (please see end of page 2 under ‘The study’).

Response: The three authors from Denmark collected the data. We have added the sentence “We collected data for the Danish arm of the international Health Behaviour in School-aged Children (HBSC) study”. This information is also available in the declaration about author contributions.

Can you reflect on the ethical implications of the questionnaire? How was consent sought? How were data stored etc? What options were provided to students who did not want to participate?

Response: We have added a text block about these issues in the Methods section.

Interesting that cyberbullying was defined/explained for the students. Was a definition of ‘loneliness’ or ‘bullying’ provided?

Response: We also defined bullying, see revised text in the Methods section. The term loneliness was not explained because prior focus group discussions with students right after they completed the questionnaire showed that the students were familiar with this concept.

Discussion

The discussion provides links with your study and the wider literature, but an exploration of the nuanced findings would add depth to the section. For example, exploring the gendered and socio-economic aspects behind loneliness or native versus non-native student’s experiences and the two types of bullying. It is possible that loneliness is related to the school environment, but these factors could also play a part.  It would be beneficial to draw on the unique results from your study rather than focusing on what they support in the literature.

Response: We agree that social contexts such as a gendered social life and socioeconomic background may be drivers of loneliness and drivers of bullying. We also agree that an exploration of these topics would be interesting. We have mentioned this issue in the revised Discussion section and proposed further studies on these issues. The focus of the present paper is however narrow: to study the association between loneliness and exposure to bullying. Dealing with social contexts such as a gendered social life and socioeconomic background these issues would work better in a separate paper.

Implications:

The paper begins with a sentence stating that bullying is subjective, have you considered the idea of research with young people to determine their understanding of this term? As you say early in the manuscript, loneliness is subjective. Therefore, does future exploration of this topic lend itself to a qualitative study exploring this issue?

Response: Thank you, this is a great idea which we have added to the section on implications for research.

Conclusion

More information is needed in the conclusion which is two sentences. What are the takeaway messages for this research? 

Response: We have followed your advice and expanded the Conclusion section with main study limitations and important implications for practice.

Reviewer 2 Report

Comments and Suggestions for Authors

Dear authors of the manuscript "Bullying at school, cyberbullying, and loneliness: National representative study of adolescents in Denmark":

I am writing to you regarding your aforementioned study, which I have had the pleasure of reviewing recently. I would like to begin by expressing my appreciation for the work you have done in this important and relevant field of research.

After a thorough review of your study, I would like to share some observations and suggestions that I believe could further improve its quality and relevance:

1. I have observed that there are different quotations that tend to be repeated in the introductory section. I think it would be appropriate to limit this repetition and look for new studies that support these ideas. 

2. It would be interesting to better conceptualize what they mean by "exposure to school violence", since they only refer to students who are victims of bullying and do not take into account other roles that are found within this problem such as observers and bullies. These roles that I expose are also exposed to situations of school violence to a greater or lesser extent. 

3. In the introduction, aspects such as the differences according to sex or age are presented very briefly, while later in the results more emphasis is placed on them. I believe that these ideas concerning sociodemographic variables should have been discussed in greater depth in the introduction. 

4. The discussion section should link the results obtained with the ideas set out in the introduction and not be limited to simply restating the results. 

5. Within the conclusions, the limitations of the study and future studies should be added.

I hope these suggestions will be helpful to you and to the continued development of your research. 

Best regards.

Author Response

Reviwer 2

Dear authors of the manuscript "Bullying at school, cyberbullying, and loneliness: National representative study of adolescents in Denmark": I am writing to you regarding your aforementioned study, which I have had the pleasure of reviewing recently. I would like to begin by expressing my appreciation for the work you have done in this important and relevant field of research.

After a thorough review of your study, I would like to share some observations and suggestions that I believe could further improve its quality and relevance:

Response: Thank you for your kind words. We have addressed all of your comments and recommendations.  

  1. I have observed that there are different quotations that tend to be repeated in the introductory section. I think it would be appropriate to limit this repetition and look for new studies that support these ideas. 

Response: We have carefully re-considered the use of references and made some changes, as you recommend. We have reduced the number of repeated references and included other references in the Discussion section which support our findings.  

  1. It would be interesting to better conceptualize what they mean by "exposure to school violence", since they only refer to students who are victims of bullying and do not take into account other roles that are found within this problem such as observers and bullies. These roles that I expose are also exposed to situations of school violence to a greater or lesser extent. 

Response: We did not introduce the term "exposure to school violence" in our manuscript because we wanted to focus on the much narrower concept, exposure to bullying at school. However, this perspective is exiting, and we have mentioned it as a relevant future research issue, see the revised Discussion section.  

  1. In the introduction, aspects such as the differences according to sex or age are presented very briefly, while later in the results more emphasis is placed on them. I believe that these ideas concerning sociodemographic variables should have been discussed in greater depth in the introduction. 

Response: We have added more text in the Introduction section to justify the focus on sex- and age differences.  

  1. The discussion section should link the results obtained with the ideas set out in the introduction and not be limited to simply restating the results. 

Response: We have revised and expanded the conclusion as you suggested.  

  1. Within the conclusions, the limitations of the study and future studies should be added.

Response: We have expanded the conclusion as you suggested.  

I hope these suggestions will be helpful to you and to the continued development of your research. 

Response: Thank you for your thorough review and constructive recommendations.

Reviewer 3 Report

Comments and Suggestions for Authors

Dear authors,

The manuscript entitled -Bullying at school, cyberbullying, and loneliness: National representative study of adolescents in Denmark- aimed to examine how loneliness was associated with school and online bullying.

The introduction provided enough motivation for creating this article.

In the materials and methods section i suggest adding a chart to better visuale the inclusion process and, also, i suggest adding the inclusion and exclusion criteria.

The results are clearly presented.

The conclusion id supported by the results.

I suggest admision of this manuscript after minor revision.

Comments on the Quality of English Language

Minor editing of English language required.

Author Response

Reviewer 3

Dear authors,

The manuscript entitled -Bullying at school, cyberbullying, and loneliness: National representative study of adolescents in Denmark- aimed to examine how loneliness was associated with school and online bullying.

The introduction provided enough motivation for creating this article.

In the materials and methods section i suggest adding a chart to better visuale the inclusion process and, also, i suggest adding the inclusion and exclusion criteria.

Response: We have added a text about the inclusion process and the inclusion and exclusion criteria. This text is, in our opinion, so instructive that a chart is not needed.

The results are clearly presented.

The conclusion id supported by the results.

I suggest admision of this manuscript after minor revision.

Response: Thank you for your kind words.

Reviewer 4 Report

Comments and Suggestions for Authors

Dear Authors:

Your work presents relevant contributions in the field of one of the problems experienced by students, bullying generates a socioemotional impact on the lives of adolescents that deserves timely attention.

I consider your work to be pertinent and correctly structured, I only have one suggestion for your consideration: 

- In the discussion it is convenient that you incorporate and compare your results with those obtained in recent research, to strengthen scientific knowledge.

Comments on the Quality of English Language

-The word "and" is repeated in line 44

poor life satisfaction [18] and mental health problems and suicidal behaviour [5,24-31]. Only a few studies explore the association.

- Change the word "differences" in line 64.

- in person bullying and loneliness and cyberbullying. In line 65: in person bullying, loneliness,  and cyberbullying

Author Response

Reviewer 4

Dear Authors:

Your work presents relevant contributions in the field of one of the problems experienced by students, bullying generates a socioemotional impact on the lives of adolescents that deserves timely attention.

I consider your work to be pertinent and correctly structured, I only have one suggestion for your consideration: 

- In the discussion it is convenient that you incorporate and compare your results with those obtained in recent research, to strengthen scientific knowledge.

Response: Thank you for your kind words. We have revised the Discussion section to meet your recommendation.